# Amyloid β-Oligomers Inhibit the Nuclear Ca^2+^ Signals and the Neuroprotective Gene Expression Induced by Gabazine in Hippocampal Neurons

**DOI:** 10.3390/antiox12111972

**Published:** 2023-11-06

**Authors:** Pedro Lobos, Ignacio Vega-Vásquez, Barbara Bruna, Silvia Gleitze, Jorge Toledo, Steffen Härtel, Cecilia Hidalgo, Andrea Paula-Lima

**Affiliations:** 1Biomedical Neuroscience Institute, Faculty of Medicine, Universidad de Chile, Santiago 8380453, Chile; ploboszqf@gmail.com (P.L.); ignaciovega@ug.uchile.cl (I.V.-V.); sgleitze@ug.uchile.cl (S.G.); shartel@uchile.cl (S.H.); 2Advanced Clinical Research Center, Clinical Hospital, Universidad de Chile, Santiago 8380456, Chile; bbruna@ug.uchile.cl (B.B.); jtoledo@uchile.cl (J.T.); 3Advanced Scientific Equipment Network (REDECA), Faculty of Medicine, Universidad de Chile, Santiago 8380453, Chile; 4Laboratory for Scientific Image Analysis, Center for Medical Informatics and Telemedicine, Faculty of Medicine, Universidad de Chile, Santiago 8380000, Chile; 5Anatomy and Biology of Development Program, Institute of Biomedical Sciences, Faculty of Medicine, Universidad de Chile, Santiago 8380000, Chile; 6Department of Neuroscience, Faculty of Medicine, Universidad de Chile, Santiago 8380000, Chile; 7Physiology and Biophysics Program, Institute of Biomedical Sciences and Center for Exercise, Metabolism and Cancer Studies, Faculty of Medicine, Universidad de Chile, Santiago 8380000, Chile; 8Interuniversity Center for Healthy Aging (CIES), Santiago 8380000, Chile; 9Institute for Research in Dental Sciences (ICOD), Faculty of Dentistry, Universidad de Chile, Santiago 8380544, Chile

**Keywords:** Alzheimer’s disease, hippocampal cultures, amyloid-beta oligomers, gabazine, neuronal activity, synaptotoxicity, calcium-dependent signaling pathways, calcium-mediated gene expression, antioxidant enzyme expression

## Abstract

Hippocampal neuronal activity generates dendritic and somatic Ca^2+^ signals, which, depending on stimulus intensity, rapidly propagate to the nucleus and induce the expression of transcription factors and genes with crucial roles in cognitive functions. Soluble amyloid-beta oligomers (AβOs), the main synaptotoxins engaged in the pathogenesis of Alzheimer’s disease, generate aberrant Ca^2+^ signals in primary hippocampal neurons, increase their oxidative tone and disrupt structural plasticity. Here, we explored the effects of sub-lethal AβOs concentrations on activity-generated nuclear Ca^2+^ signals and on the Ca^2+^-dependent expression of neuroprotective genes. To induce neuronal activity, neuron-enriched primary hippocampal cultures were treated with the GABA_A_ receptor blocker gabazine (GBZ), and nuclear Ca^2+^ signals were measured in AβOs-treated or control neurons transfected with a genetically encoded nuclear Ca^2+^ sensor. Incubation (6 h) with AβOs significantly reduced the nuclear Ca^2+^ signals and the enhanced phosphorylation of cyclic AMP response element-binding protein (CREB) induced by GBZ. Likewise, incubation (6 h) with AβOs significantly reduced the GBZ-induced increases in the mRNA levels of neuronal Per-Arnt-Sim domain protein 4 (Npas4), brain-derived neurotrophic factor (BDNF), ryanodine receptor type-2 (RyR2), and the antioxidant enzyme NADPH-quinone oxidoreductase (Nqo1). Based on these findings we propose that AβOs, by inhibiting the generation of activity-induced nuclear Ca^2+^ signals, disrupt key neuroprotective gene expression pathways required for hippocampal-dependent learning and memory processes.

## 1. Introduction

Alzheimer’s disease (AD), the most common form of age-associated dementia worldwide, is a progressive and fatal neurodegenerative illness manifested by severe and progressive deterioration of cognitive and memory processes [1]. Extracellular senile plaques, composed mainly of β-amyloid (Aβ) peptides [2], and intracellular neurofibrillary tangles of hyperphosphorylated microtubule-associated protein (p-Tau) [3,4], are the primary brain histopathological AD markers. At early stages in this disease, Aβ and p-Tau aggregates accumulate in the hippocampus, a brain region with an essential role in learning and memory processes [4], and which is particularly vulnerable to the neurotoxic effects associated with different age-related diseases [5,6,7]. Hippocampal atrophy is an early indication of the conversion from a normal aging process to the development of dementia [8,9]. Therefore, AβOs treatment of primary hippocampal neurons is a suitable experimental in vitro model with which to study the mechanisms underlying the neuronal dysfunctions promoted by AβOs [10,11,12,13,14,15].

Uncontrolled neuronal activity, as occurs in AD, affects neural network homeostasis and initiates synaptotoxic processes [16]. Current evidence indicates that multiple factors contribute to the development of AD [17]; among them, neuronal oxidative stress and abnormal Ca^2+^ signaling are central features of this disease [18,19]. Several studies have shown that postmortem AD brains display elevated oxidative stress markers, including oxidized lipids, proteins, and DNA (reviewed in [20]). Even slight imbalances of these molecules can be deleterious to the brain; moreover, elevated neuronal levels of reactive oxygen species (ROS) can cause selective dysfunctions and neurodegeneration [21,22,23]. Memory mechanisms are directly compromised by elevated ROS content, which underscores the importance of establishing how excessive ROS levels may be coupled with other critical aspects of AD pathology. The first link between AβOs and neuronal oxidative stress was established in 2007 [11] and was confirmed by us and other groups [15,24].

Neuronal Ca^2+^ signals, defined as transient, controlled increments in intracellular Ca^2+^ concentration, play essential roles in hippocampal synaptic plasticity and memory processes [25,26]. However, uncontrolled Ca^2+^ signals lead to neuronal dysfunction and eventually cause neuronal death [27]. Accordingly, neurons have developed highly sophisticated mechanisms to control Ca^2+^ homeostasis and signaling, locally in dendritic spines, and in somatic and nuclear compartments [25,28]; these control mechanisms fail to function properly in AD [19,29]. In primary hippocampal cultures, AβOs promote Ca^2+^ influx through N-methyl-D-aspartate (NMDA) glutamate receptors, which induces Ca^2+^ release mediated by ryanodine receptor (RyR) Ca^2+^ channels [12]. The ensuing abnormal Ca^2+^ signals, which are long-lasting but of low amplitude, engage Ca^2+^-dependent pathways that hinder BDNF-induced dendritic spine growth and induce excitotoxicity, leading to the loss of synaptic structure and function [12,29,30]. Of note, it was recently shown in hippocampal neurons that steep increases in nuclear Ca^2^⁺ levels induce instantaneous uncoupling of a protein called Jacob from LaminB1 at the nuclear lamina and promote the association of Jacob with the transcription factor cAMP-responsive element-binding protein (CREB) [31]. Interestingly, transient receptor potential V1 (TRPV1) deficiency, by promoting the BDNF/TrkB signaling pathway, prevents hippocampal cell death in 3xTg-AD mice [32]. In summary, abnormal intracellular Ca^2+^ signaling and the related excitable and synaptic dysfunctions are consolidated hallmarks of AD onset and possibly of other neurodegenerative diseases [33,34].

Essentially, all activity-induced, long-term modifications of brain functions require appropriate synapse-to-nucleus communication to precisely control nuclear Ca^2+^-dependent gene expression [28]. Robust synaptic activity generates Ca^2+^ signals that, in addition to promoting the activation of cytoplasmic signaling cascades, readily reach and enter the nucleus via passive diffusion through nuclear pores [35,36]. However, nuclear Ca^2+^ entry requires Ca^2+^ signals to reach the vicinity of the nucleus, which does not occur by simple Ca^2+^ diffusion since this process is highly restricted in the cytoplasm [37,38]. The type-2 (RyR2) RyR isoform plays a major role in the propagation of Ca^2+^ signals to the nucleus [26]. Once inside the nucleus, Ca^2+^ signals contribute to activity-dependent transcription by promoting Ca^2+^-dependent phosphorylation of CREB-binding protein (CBP), which allows the formation of the transcriptionally active CREB-CBP complex [28,39,40].

Blocking GABA_A_ receptor function with gabazine (GBZ) stimulates the activity of rat primary hippocampal neurons [26,41,42], which contain ~10–15% of the inhibitory interneurons responsible for the tonic inhibition of the neuronal network [43]. The GBZ-induced increase in hippocampal neuronal activity promotes the emergence of oscillatory Ca^2+^ transients, which propagate from the cytoplasm to the neuronal nucleus, where they promote CREB phosphorylation [26]—a key event in BDNF-mediated synaptic plasticity and hippocampal memory [44,45]. The addition of GBZ also enhances BDNF, Npas4, and RyR2 expression [26], all of which have central roles in hippocampal synaptic plasticity and spatial memory [45,46,47,48]. In contrast, the incubation (6 h) of primary hippocampal neurons with a sub-lethal AβOs concentration eliminates the RyR2 protein increase induced by BDNF and prevents, within minutes, the dendritic spine remodeling induced by BDNF [12]. Moreover, the incubation (6 h) of primary hippocampal neurons with AβOs inhibits the RyR2-mediated generation of mitochondrial ROS, and thus contributes to the synaptic dysfunction caused by AβOs [15]. However, the effects of AβOs on the generation of activity-induced nuclear Ca^2+^ signals, synaptic plasticity, and antioxidant defense mechanisms in hippocampal neurons remain undefined.

In this work, we report that AβOs inhibited the generation of nuclear Ca^2+^ signals induced by GBZ addition to rat primary hippocampal neurons; AβOs also decreased the GBZ-induced CREB phosphorylation increase, and the expression of genes engaged in memory and neuroprotective processes, such as BDNF, RyR2, and Npas4. Importantly, AβOs also reduced the expression of NADPH-quinone oxidoreductase-1 (Nqo1), an enzyme with a key role in the neuronal antioxidant response that is significantly impaired in AD [49,50]. In this study, by revealing that AβOs disrupt crucial nuclear Ca^2+^-signaling processes and suppress the expression of genes associated with neuroprotection in primary hippocampal cultures, we present novel results that advance our current knowledge of the noxious effects of AβOs on rodent hippocampal function. Hence, this work provides a framework for a deeper understanding of how AD-related mechanisms impact synaptic plasticity and neuronal health. Moreover, it offers promising avenues for therapeutic interventions in AD and related neurodegenerative conditions.

## 2. Materials and Methods

**Reagents:** The Aβ1–42 peptide was from Bachem Inc. (Torrance, CA, USA), gabazine SR95531 was from Tocris Bioscience (Avonmouth, Bristol, UK); cytoplasmic GCaMP5 and nuclear GCaMP3-NLS were kindly provided by Dr. H. Bading, University of Heidelberg, Germany; DMSO was from Sigma (HE, Darmstadt, Germany), and hexafluoro-2-propanol (HFIP) was from Merck (HE, Darmstadt, Germany). The serum-free neurobasal medium, B27 supplement, GlutaMAX™, DAPI, Lipofectamine 2000 and TRIzol reagent were from Gibco™/Thermofisher Scientific (Waltham, MA, USA). Brilliant III Ultra-Fast SYBR^®^ Green QPCR Master Mix and DAKO mounting medium were from Agilent Technologies (Santa Clara, CA, USA). The DNAase Turbo DNA-freeTM kit was from Ambion™/Thermofisher Scientific (Waltham, MA, USA). Improm II TM reverse transcriptase was from Promega (Madison, WI, USA). The protein content was determined with a BCA kit from Thermofisher Scientific (Waltham, MA, USA).

**Antibodies:** Specific antibodies against p-CREB were from Cell Signaling Technologies (Danvers, MA, USA). Antibodies against MAP2 were from Abcam (Cambridge, UK) and those against β-tubulin were from Sigma-Aldrich (St. Louis, MO, USA). Alexa Fluor 488 anti-rabbit and Alexa Fluor 635 anti-mouse were from Invitrogen (Carlsbad, CA, USA).

**Primary rat hippocampal cultures:** Primary hippocampal cultures were prepared from Sprague Dawley rats at a gestation age of 18 days, as described in [12]. Briefly, cells were recovered and plated in Petri dishes previously treated with poli-L-lysine (0.1 mg/mL). Then, cultures were maintained in a serum-free neurobasal medium containing B27 supplement, glutamax TM, penicillin (20 U/mL)/streptomycin (20 μg/mL) at 37 °C under 5% CO_2_. In all experiments, cultures were used between 12–15 days in vitro (DIV). For most of the experiments, hippocampal neurons were pre-incubated with 500 nM AβOs for 6 h before incubation with 5 μM gabazine for 30 min for Ca^2+^ measurements and CREB immunofluorescence; for determinations by qPCR, cultures were incubated for 6 h with 500 nM AβOs before incubation with 5 μM gabazine for 2 h. Compounds were maintained during the respective incubation period, as stated. All experimental protocols used in this work complied with the “Guiding Principles for Research Involving Animals and Human Beings” of the American Physiological Society and were approved by the Bioethics Committee on Animal Research, Faculty of Medicine, Universidad de Chile.

**Preparation of AβOs:** The Aβ1–42 peptide was prepared as a dried hexafluoro-2-propanol (HIFP) film and stored at −20 °C, as described in [12,13,15,24]. Before use, the film was resuspended in DMSO, diluted with 100 μM cold phosphate-buffered saline (PBS) and incubated without stirring for 24 h at 4 °C. The resulting Aβ-containing solution was centrifuged at 14,000× *g* for 10 min at 4 °C to remove protofibrils and fibrils (insoluble aggregates). The supernatants that contained Aβ oligomers were transferred to clean tubes and stored at 4 °C; an aliquot was used to determine AβOs content with a BCA kit. Fresh AβOs preparations (up to 48 h) were used in all experiments.

**Cytoplasmic Ca^2+^ Signals:** To detect cytoplasmic Ca^2+^ signals, primary hippocampal cultures were transfected with the genetically encoded cytoplasmic Ca^2+^ sensor GCaMP5. Transfected neurons were recorded in Tyrode’s solution (mM: 129 NaCl, 5 KCl, 25 HEPES, pH 7.3, 30 glucose, 2 CaCl_2_, and 1 MgCl_2_). As an additional strategy to detect neuronal Ca^2+^ signals, primary cultures were loaded for 15 min at 37 °C with the Ca^2+^ probe Fluo 4-AM, prepared in Tyrode’s extracellular medium. Cells were placed in the microscopy stage of a wide-field Zeiss Cell Observer epifluorescence microscope (Zeiss, Oberkochen, Germany), using, as objectives, Plan-Neofluar 20×/0.4 or Plan Apochromat, 40×/1.3 water, with a light source 470 nm Colibri 2 LED-based module and a digital camera EMCCD Evolve 512 delta (Teledyne Photometrics, Tucson, AZ, USA). All settings were adjusted to minimize bleaching and maximize acquisition frequency. After recording a stable baseline, 10 μM of gabazine was added to the culture; ionomycin (3 μM) was added at the end of the experiment to visualize the maximum fluorescence signals.

To determine basal Ca^2+^ levels, cultures were incubated at 37 °C for 20 min in the dark with 2 µM Fura-2 AM. Subsequently, cells were washed three times with Tyrode’s solution (mM: 129 NaCl, 5 KCl, 25 Hepes, pH 7.3, 30 glucose, 2 CaCl_2_, 1 MgCl_2_). Cells were placed in the microscopy stage of a Spinning Disc IX81 microscope (Olympus, Tokyo, Japan), using a xenon lamp as the excitation source, with excitation filters for 340/402 nm wavelengths, and a 40× objective. The first minutes of the record yielded the fluorescence levels corresponding to the resting Ca^2+^ concentration. Then, ionomycin was added to saturate the Ca^2+^ probe. For data analysis, the background fluorescence was subtracted and the ratio of the 340 nm and 380 nm fluorescence intensity (ratio; F_340_/F_380_) was calculated.

**Nuclear Ca^2+^ Signals:** To detect nuclear Ca^2+^ signals, primary hippocampal cultures (14–16 DIV) were transfected with the genetically encoded Ca^2+^ sensor GCaMP3-NLS, which has a sequence of nuclear destination and thus accumulates in the neuronal nucleus. The GCaMP3-transfected neurons were recorded in Tyrode’s solution (mM: 129 NaCl, 5 KCl, 25 Hepes, pH 7.3, 30 glucose, 2 CaCl_2_, 1 MgCl_2_). To assess the direct effect of AβOs on nuclear Ca^2+^ signals, cultures were incubated at the microscope stage with 500 nM AβOs and acquired by image time-lapse for 15 min. To determine the effect of AβOs on activity-induced nuclear Ca^2+^ signals, neurons were incubated for 6 h with 500 nM AβOs before the addition of 5 μM GBZ at the microscope stage. A wide-field Zeiss Cell Observer epifluorescence microscope (Zeiss, Jena, Germany), using Plan-Neofluar 20 Å~/0.4 or Plan Apochromat as objectives, 40 Å~/1.3 water, with a light source, 470-nm Colibri 2 light-emitting diode (LED)-based module, and a digital camera, an electron-multiplying charge-coupled device (EMCCD) Evolve 512 delta (Teledyne Photometrics, Tucson, AZ, USA) was used. All settings were adjusted to minimize bleaching and maximize acquisition frequency. After recording a stable baseline, 5 μM GBZ was added to the culture; 3 μM ionomycin was added at the end of the experiment to visualize the maximum fluorescence signals. The analysis of Ca^2+^ signals was performed as described in [26]. Briefly, images were segmented by hand to generate binary masks suitable for morphological and functional analysis using the Fiji distribution of ImageJ 1.54f software. All fluorescence signals, recorded at the region of stimulation, are expressed as (F_max_ − F_0_/F_0_) or as F/F_0_, where F_max_ and F represent, respectively, the maximal recorded fluorescence intensity and the recorded fluorescence intensity; F_0_ corresponds to the intensity at the initial time (mean intensity of 20 to 50 frames recorded before the stimulus).

**CREB phosphorylation and nuclear staining:** To determine whether AβOs interrupt the GBZ-induced increase in synaptic activity and CREB phosphorylation, hippocampal cultures were pre-incubated for 6 h with 500 nM AβOs before the addition of 5 μM GBZ for 30 min. Cells were then fixed and CREB phosphorylation was detected by immunofluorescence using an antibody against the phosphorylated form of CREB (Ser-133 p-CREB). In addition, cultures were probed with an antibody against the microtubule-associated protein MAP2 to identify neuronal cells and with DAPI to detect all cell nuclei. All images were recorded with a 40× objective in the z-axis of all confocal planes acquired by spinning disk microscopy (Olympus XI 81 Spinning Disk super zoom, Tokyo, Japan). Quantification of the relative levels of p-CREB fluorescence was detected in different conditions by background subtraction (50 pixels ball radius) and segmentation by Otsu threshold, using DAPI staining as the total nuclear staining reference. Mean fluorescence intensity was quantified only in neuronal cells (MAP2 positive) and expressed as a fold change relative to the control cells.

**RNA Extraction and qPCR:** Total RNA was isolated using TRIzol reagent following the manufacturer’s recommendations; to remove any contamination with genomic DNA; a digestion step with DNAase (Turbo DNA-freeTM kit, Ambion, Invitrogen, Carlsbad, CA, USA) was included to remove residual contaminating DNA. The RNA purity was determined by the 260/280 absorbance ratio in nanodrop (Nano-400 Nucleic Acid Analyzer from Allsheng (Hangzhou, China). Complementary DNA (cDNA) was synthesized with Improm II TM reverse transcriptase using 2 μg of total RNA. The mRNA levels of RyR2, BDNF exon IV, Npas4, and NqoI were amplified using specific primers (Appendix A) from 2 μg of cDNA in 20 μL final volume and were performed in an Mx3000P qPCR System (Stratagene, Santa Clara, CA, USA) using the Brilliant III Ultra-Fast SYBR^®^ Green QPCR Master Mix. The mRNA levels were quantified with the 2^−ΔΔCT^ method [51] using β-actin as the housekeeping gene. All samples were run in triplicate and included controls. To verify the purity of the products, the dissociation curves were analyzed in all cases.

**Statistical and Data Analysis:** Results are expressed as mean ± SE. Statistical analysis between groups was performed with one-way ANOVA followed by the Bonferroni test, as indicated in the respective figure legends. The comparison between the two groups was performed by two-tailed Student’s *t*-test. All statistical analyses were performed using SigmaPlot version 12.0.

## 3. Results

### 3.1. AβOs Disrupt the Nuclear Ca^2+^ Transients Induced by Gabazine

Previous reports have described that the addition of AβOs elicits long-lasting and low-amplitude cytoplasmic Ca^2+^ signals in primary hippocampal neurons [12,13]. Following AβOs addition (500 nM), cytoplasmic Ca^2+^ signals, detected with Fluo-4 (Figure 1A), exhibited a similar response pattern (Figure 1B). To evaluate if the AβOs also induced nuclear Ca^2+^ signals, AβOs (500 nM) were added at the microscope stage to hippocampal cultures transfected with the nuclear Ca^2+^ sensor GCaMP3-NLS (Figure 1C). This treatment induced a slow, sustained, and low-intensity increase in nuclear Ca^2+^ levels (Figure 1D). This nuclear response was slower than that displayed by the cytoplasmic Ca^2+^ signals following AβOs addition to primary hippocampal neurons (Figure 1B) [12,15,24]. The results are presented as representative recordings from experiments performed in different hippocampal cultures; *n* = 3.

To test the effects of AβOs on the Ca^2+^ signals induced by neuronal activity, primary cultures preincubated (6 h) with AβOs were treated with the GABA_A_ receptor antagonist GBZ, which, by decreasing the inhibitory GABAergic tone, produces an excitatory/inhibitory imbalance and a rapid increase in neuronal activity [41,52]. As reported previously, treatment of primary hippocampal cultures with GBZ induced the emergence of oscillatory and synchronized cytoplasmic (Figure 2A,B) and nuclear Ca^2+^ signals (Figure 2C,D), measured as described in [26]. The GBZ-induced nuclear Ca^2+^ transients, recorded in primary neurons transfected with the genetically encoded nuclear Ca^2+^ sensor GCaMP3-NLS (Figure 2C), displayed frequencies, intensities, and durations that exhibited some experimental variability (Figure 2D). Data are expressed as representative recordings from experiments performed with different hippocampal cultures; *n* = 5 (and 50 neurons analyzed per condition).

In contrast, primary neurons pre-treated (6 h) with 500 nM AβOs before GBZ addition displayed a conspicuous decrease in the emergence of the oscillatory and synchronic nuclear Ca^2+^ signals elicited by GBZ (Appendix A). Primary cultures were incubated for 6 h with 500 nM AβOs since this protocol significantly alters hippocampal function by deregulating glutamate neurotransmission, leading to synapse failure [29]. As depicted in Appendix A, 15 min incubation with AβOs did not inhibit the induction of nuclear Ca^2+^ signals by GBZ. Quantification of different nuclear Ca^2+^ signals parameters revealed that AβOs treatment for 6 h significantly decreased the maximum amplitude (Figure 3A) and the number of detected peaks (Figure 3B) induced by GBZ addition. Data are expressed as Mean ± SE; *n* = 5 of independent experiments (different primary cultures); at least 50 neurons in total were analyzed per condition. Statistical analysis was performed with Student’s *t*-test. *, the exact p values were *p* = 0.040 in Figure 3A and *p* = 0.037 in Figure 3B. These results show that incubation (6 h) with 500 nM AβOs, which is a non-lethal concentration [12,53], disrupts the oscillatory and synchronic nuclear Ca^2+^ signals caused by the increase in neuronal activity induced by GBZ. Of note, treatment with 500 nM AβOs (6 h) did not modify the basal cytoplasmic Ca^2+^ levels (Appendix A). These results indicate that the moderate increase in cytoplasmic Ca^2+^ levels initially caused by AβOs, which lasts for at least 30 min [26], did not persist in primary hippocampal neurons incubated for 6 h with 500 nM AβOs.

### 3.2. AβOs Prevent the CREB Phosphorylation Increase Induced by Gabazine

The increase in hippocampal neuronal activity induced by GBZ triggers the early induction of gene expression changes commanded initially, among other factors, by the phosphorylation of the transcription factor CREB and of its binding protein CBP by different nuclear kinases [26,28]. To determine whether treatment with AβOs disrupts the CREB phosphorylation increase induced by GBZ, primary cultures were incubated for 6 h with 500 nM AβOs prior to GBZ addition. Phosphorylated CREB (p-CREB) levels were detected by immunofluorescence 30 min after GBZ addition, using an antibody against CREB phosphorylated in serine 133 (green in Figure 4). An antibody against the microtubule-associated protein MAP2 was used to identify neurons (red in Figure 4) and DAPI was applied to detect all cell nuclei (blue in Figure 4). Treatment with GBZ for 30 min produced a significant increase in CREB phosphorylation in primary neurons (Figure 4A,B), as described in [26]. In contrast, neurons preincubated for 6 h with 500 nM AβOs and then incubated with 5 µM GBZ for 30 min showed significantly lower p-CREB levels, which were not different from those displayed by the untreated controls (Figure 4B). Bars represent Mean ± SE, *n* = 3. **: *p* < 0.01 Statistically significant differences were evaluated by one-way ANOVA followed by Bonferroni’s post hoc test for multiple comparisons. **: *p* < 0.01. From left to right, the exact *p* values are *p* = 0.003; *p* = 0.030 and *p* = 0.005. These results show that incubation for 6 h with AβOs abolished the subsequent GBZ-induced CREB phosphorylation increase.

### 3.3. AβOs Inhibit the Increase in Npas4, RyR2, BDNF and Nqo1 mRNA Levels Induced by Gabazine

The increase in neuronal activity induced by GBZ, together with promoting CREB phosphorylation also increases the expression of the early-response gene Npas4 [26]. In agreement, 2 h treatment of hippocampal cultures with 5 µM GBZ induced a significant increase in the mRNA levels of Npas4 (Figure 5A). The addition of GBZ also increased RyR2 mRNA levels (Figure 5B), and triggered a neuroprotective response, evidenced by significant increases in the mRNA levels of exon IV of the neurotrophic factor BDNF (Figure 5C). Of note, GBZ also promoted a significant increase in the mRNA levels of the antioxidant enzyme Nqo1 (Figure 5D). However, neurons treated with 500 nM AβOs for 6 h before addition of 5 µM GBZ displayed, after 30 min, significantly lower mRNA levels of Npas4 (Figure 5A) and BDNF exon IV (Figure 5C), whereas the RyR2 and Nqo1 mRNA levels were reduced to the control values (Figure 5B,D). Data are expressed as Mean ± SE; *n* = 5. Statistical analysis was performed with ANOVA followed by Tukey’s post hoc test for multiple comparisons. *: *p* < 0.05; **: *p* < 0.01. From left to right, the exact *p* values are *p* = 0.0002 and *p* = 0.0311 for Figure 5A; *p* = 0.009; *p* = 0.013 and *p* = 0.025 for Figure 5B; *p* < 0.001; *p* = 0.039 for Figure 5C; *p* = 0.013 and *p* = 0.039 for Figure 5D.

Taken together, these results show that incubation with AβOs inhibited the nuclear Ca^2+^ signals induced by neuronal activity and reduced or abolished the GBZ-induced increases in CREB phosphorylation and of Npas4, RyR2, BDNF, and Nqo1 mRNA levels.

## 4. Discussion

Alterations in synapse-to-nucleus communication, which occur in different central nervous system disorders, contribute to the development of neurodegenerative diseases [34]. The results presented here show that treatment for 6 h of primary hippocampal neurons with AβOs decreased the production of GBZ-induced nuclear Ca^2+^ signals and the phosphorylation of the transcription factor CREB, and reduced or abolished the enhanced expression of Npas4, BDNF, RyR2, and Nqo1 induced by GBZ.

Neuronal Ca^2+^ signals have a wide range of functions in the activity-dependent synaptic plasticity processes that underlie hippocampal spatial memory [54,55,56]. Among the molecular players controlling neuronal Ca^2+^ signaling, the endoplasmic reticulum resident inositol 1,4,5-trisphosphate receptors and RyR channels have central roles. Both Ca^2+^ channel types amplify Ca^2+^ entry signals via Ca^2+^-induced Ca^2+^ release and thus contribute to regulating the function of neuronal Ca^2+^-dependent signaling pathways [24,57,58,59,60]. Of note, RyR-mediated Ca^2+^ release plays an essential role in hippocampal synaptic plasticity, and the formation/consolidation of spatial memory [45,46,61,62]. Changes in the cellular oxidative state particularly affect RyR function, so that reducing agents impede RyR activation by Ca^2+^, whereas oxidizing agents have the opposite effect [24,57,58,59,60]. Excessive ROS-induced RyR channel activation has been associated with abnormal Ca^2+^ release and the functional defects associated with AD and age-related hippocampal dysfunction [12,15,24,46,63,64,65,66].

In this work, to induce neuronal activity, primary hippocampal cultures were treated with 5 µM GBZ, which at this concentration effectively inhibits GABA_A_ receptors [26,52,67]; moreover, GBZ is a more potent and specific GABA_A_ receptor antagonist than bicuculline, pentobarbital, and alfaxalone [52,68]. Additionally, we recently reported [26] that the addition of 5–10 µM GBZ to primary hippocampal neurons induces neuronal activity, promotes the emergence of cytoplasmic and nuclear calcium signals, and increases CREB phosphorylation and the expression of genes involved in synaptic plasticity and neuroprotection.

Previous studies have reported that AβOs are synaptotoxins that lead to cognitive impairment in AD [69] and impede synaptic plasticity by inhibiting long-term potentiation (LTP), enhancing long-term depression (LTD) [70]; AβOs also impair structural plasticity by reducing dendritic spine density in normal rodent hippocampus and primary hippocampal neurons [12,70]. Electrophysiological recordings coupled with the expression of synaptic genes would provide valuable insights into the mechanisms of AβOs-invoked disruption of synaptic plasticity induced by GBZ. Our findings on the inhibition of Ca^2+^ signaling produced by AβOs in primary hippocampal neurons provide a foundation for future research exploring synaptic plasticity and electrophysiological aspects in more detail.

Our in vitro rat model of AβOs associated synaptotoxicity, which entails acute treatment of primary hippocampal cultures with non-lethal AβOs concentrations [12], adequately represents the deleterious changes in Ca^2+^ signaling that occur during aging and in response to Aβ peptide toxicity [71]. In agreement with these findings, a recent study reported that iPSC differentiated from the neurons of patients with presenilin-1 (PS1) mutations display Ca^2+^ homeostasis failures and increased β-amyloid and p-Tau levels; negative allosteric modulators of RyR channels reverse these effects, reaffirming the importance of regulating RyR-mediated Ca^2+^ release in AD-affected neurons [72]. Additional studies, performed in a triple transgenic AD model, have confirmed the relevance of controlling RyR-mediated Ca^2+^ release for neutralizing the AD disease phenotype [73]. Moreover, studies in elderly Rhesus macaques have reported that RyR-mediated Ca^2+^ release enhances Tau phosphorylation and is associated with reduced neuronal activation and cognitive impairment [63]. Additionally, RyR2 isoform overactivation induces neuronal hyperactivity and Aβ accumulation in a feedback cycle that triggers dendritic spine loss, decreased memory, and death; the negative RyR2 modulator R-Carvedilol and RyR2 mutations aimed at decreasing its activity prevent these processes in humans and mice [64,74].

Adding to these studies, the present results show that acute AβOs treatment inhibited the generation of nuclear Ca^2+^ signals induced by neuronal activity, a process that engages RyR2-mediated Ca^2+^ release [26]. Of note, acute AβOs treatment of primary hippocampal neurons induces the generation of sustained, low-magnitude cytoplasmic Ca^2+^ signals that are RyR-mediated [12]. As reported here, the same pattern of Ca^2+^ signals were generated by AβOs in the neuronal nucleus. Interestingly, basal cytoplasmic Ca^2+^ levels, as determined by the ratiometric fluorescence indicator Fura-2, were not significantly different between control neurons and neurons treated for 6 h with AβOs. These results indicate that primary hippocampal neurons can restore, with time, cytoplasmic Ca^2+^ levels within a normal range, despite the low-intensity and long-lasting Ca^2+^ increases that AβOs promote for at least the first 30 min in the cytoplasm. Nevertheless, by mechanisms that are not fully elucidated, incubation for 6 h with AβOs significantly reduced the nuclear Ca^2+^ signal generation induced by GBZ. Hence, neuronal stimulation seems to potentiate the inhibitory effects of AβOs on Ca^2+^ dynamics, particularly on nuclear Ca^2+^ signal generation, a result which may have implications for the deleterious effects of AβOs on hippocampal neuronal function during the performance of tasks that entail neuronal activation and engage RyR-mediated Ca^2+^ release. The development of pharmacological strategies that modulate the activity of RyR channels will continue to attract interest in the development of new drugs to regulate neurodegenerative processes [75].

The inhibitory effects of AβOs on the enhanced expression of Npas4 triggered by GBZ-induced neuronal activity reported here may contribute to understand the novel and central role of Npas4 in the context of AD. The immediate–early gene Npas4 is expressed only in neurons and is considered as a molecular link between neuronal activity and memory [76]. Moreover, Npas4 is among the most rapidly induced early genes, and its expression, which requires nuclear Ca^2+^ signals, is selectively induced by neuronal activity [77,78]. By orchestrating distinct activity-dependent gene programs in different neuronal populations, the transcription factor Npas4 affects synaptic connections in excitatory and inhibitory neurons, neural circuit plasticity, memory formation, reward-related gene expression and behavior, and has been involved in neuroprotection [76]. Recent studies have proposed a close relationship between the pathological processes that cause AD and the activation of pathways related to neuronal activity. Among them, Npas4 displays decreased expression in AD patients, which is related to increased aggregation and deposition of Tau protein in neurofibrillary tangles, one of the hallmarks of AD progression [79]. The decreased Npas4 expression reported in AD patients may have additional harmful effects, since Npas4 facilitates autophagy-mediated Tau elimination [80]. In contrast, the amyloid precursor protein (APP), through the generation of an intracellular (AICD) domain, controls Npas4 expression, helping to maintain excitatory/inhibitory tone balance [81]. In the same vein, a novel activity-dependent DNA repair mechanism involving the Npas4–NuA4 complex was recently uncovered; this novel mechanism has an important role in maintaining genome stability, a process that is disrupted during pathological aging [82].

Moreover, our results, which indicate that AβOs reduced the increase in Npas4 gene expression induced by gabazine in primary hippocampal neurons, open a new direction for extensive future research, including neuroinflammation. Of note, neuroinflammation, which is closely related to neurodegenerative processes, also seems to play an important role in the regulation of the excitatory/inhibitory tone; infiltration of CD8+ T lymphocytes into the hippocampus of a transgenic AD model (APP-PS1) alters the expression of Npas4 and Arc along with other genes related to synaptic plasticity and Ca^2+^ signaling [83]. Moreover, HDAC3 maintains basal epigenetic repression of Npas4 and BDNF transcription, highlighting the role of epigenetic modifications in the control of neuronal activity and induction of immediate early genes [84,85].

Altogether, these studies position Npas4 and BDNF as outstanding therapeutic targets against AβOs-induced synaptotoxicity and may constitute targets to prevent the development of AD and other neurodegenerative diseases [86,87,88]. In addition, strategies to control intracellular ROS- and Ca^2+^-mediated signaling might prevent uncontrolled activation of Npas4 and neurodegenerative processes [89].

Many studies have established that BDNF, a neurotrophin synthesized and released from neurons in an activity-dependent manner [90], mediates the morphological changes entailing the generation and growth of dendritic spines during synaptic plasticity [91,92,93]. Upon binding to TrkB receptors, BDNF stimulates several intracellular signaling cascades that require RyR-mediated Ca^2+^ release [45], including Ca^2+^-dependent kinase pathways that contribute to inducing and maintaining hippocampal LTP [94]. In this work, we focused on the expression of the BDNF exon 4 transcript because it encodes the mature BDNF protein, it is broadly expressed and strongly induced by neuronal activity in cortical and hippocampal neurons, with a tight temporal, spatial, and stimulus-specific regulation compared to mRNA expression of other BDNF transcripts like exon 3; of special interest, several calcium-response elements are found within the BDNF promoter IV region, and CREB, in combination with its coactivators CBP and CRTC1, specifically regulate its early induction in hippocampal neurons [95,96]. In view of the key roles played by both RyR and BDNF in synaptic plasticity [92,94,97,98,99], our findings that AβOs interrupt GBZ-induced nuclear Ca^2+^ signals and the ensuing expression of both BDNF and RyR2 mRNA levels, may add to our current understanding of AD pathology. In particular, the AβOs-induced decrease of BDNF mRNA levels induced by neuronal activity may compromise BDNF-induced intracellular signaling pathways relevant to hippocampal function [12,53,100,101].

Of note, coupled with an increase in neuronal activity, BDNF induces the nuclear translocation of the transcription factor Nrf2 in neurons [102], a master regulator of antioxidant protein expression that protects brain cells against oxidative damage [103,104,105]. It has been proposed that the key role played by BDNF as an inducer of neuronal antioxidant responses entails crosstalk between RyR-mediated Ca^2+^ release and ROS [102]. One of the targets of Nrf2 signaling is the expression of Nqo1, an oxidoreductase that catalyzes the 2-electron reduction of quinones to hydroquinones, which is a cellular detoxification process [106]. Here, we show that the activity dependent increase in BDNF expression mediated by GBZ is accompanied by an increase in the expression of Nqo1. Moreover, pre-treatment of hippocampal neurons with AβOs prevented the nuclear Ca^2+^-dependent expression of Nqo1 induced by GBZ, indicating that these aggregates disrupt neuroprotective pathways induced by neuronal activity that regulate neuronal oxidative tone.

## 5. Conclusions

Based on the results of this study, which provides the first integrative evidence that AβOs disrupt the generation of activity dependent nuclear Ca^2+^ signals and reduce the expression of genes engaged in synaptic plasticity and the antioxidant response, we propose that these disruptions are likely to contribute to the cognitive decline and the neurodegeneration that characterize Alzheimer’s disease. Moreover, this work adds to the current understanding of the toxic effects of AβOs on hippocampal neuronal function and reveals new possible therapeutic targets to treat AD. Hence, one potential therapeutic approach to slow or prevent AD progression would be to restore activity-induced nuclear Ca^2+^ signaling and early gene expression. More research is needed to develop and test these therapeutic approaches. Of note, a recent report analyzes how AβOs can become potential therapeutic targets for AD [107]. Recently, several clinical trials using monoclonal antibodies that target amyloid plaques or AβOs in early AD have shown positive results and even obtained approval, although with important clinical efficacy and safety issues [108,109,110]. In order to overcome these problems, it is crucial to first gain a deeper understanding of the mechanisms that come into play and ultimately disrupt the regulation of neuronal activity induced by AβOs, with significant implications for tackling the heterogeneity of this devasting pathology and for developing diagnoses and novel pharmacological targets.

## Figures and Tables

**Figure 1 antioxidants-12-01972-f001:**
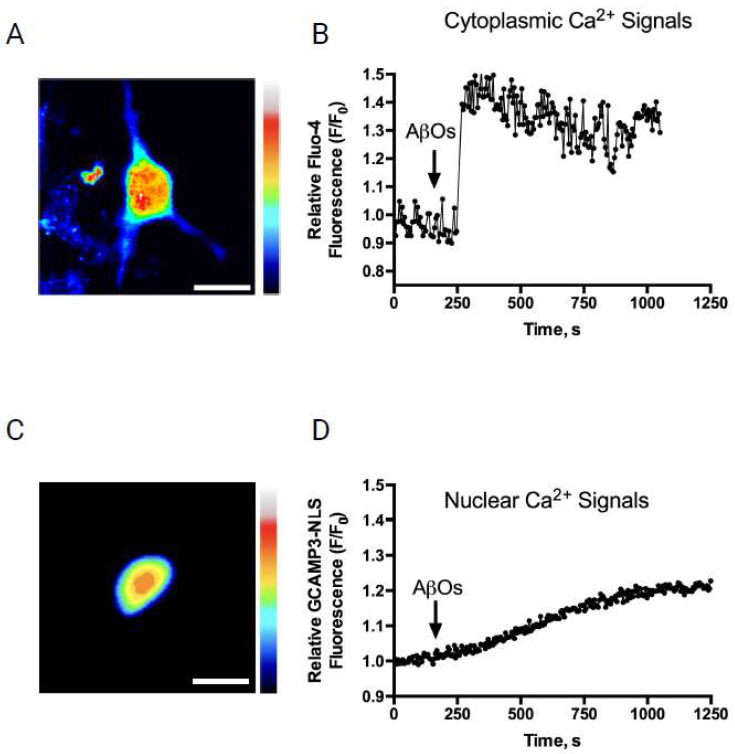
AβOs addition induces fast cytoplasmic and slower nuclear Ca^2+^ signals in pyramidal hippocampal neurons. AβOs addition induces fast cytoplasmic and slower nuclear Ca^2+^ levels in pyramidal hippocampal neurons: (**A**) representative image showing the cytoplasmic fluorescence signals emitted by a hippocampal neuron loaded with Fluo-4 after treatment with 500 nM AβOs, at the maximum level of intensity recorded. Scale bar, 15 μm. A rainbow scale was used to show the fluorescence intensity of the Ca^2+^ indicator (black: low values; red: high values); (**B**) representative trace of changes over time in cytoplasmic Ca^2+^ levels recorded in a neuron loaded with Fluo-4, before (baseline) and after the addition of 500 nM AβOs at the indicated time (arrow); (**C**) representative image showing the nuclear fluorescence signals emitted by the nucleus of a neuron transfected with GCaMP3-NLS after the addition of 500 nM AβOs to primary hippocampal cultures, at the maximum level of intensity recorded. Scale bar, 20 μm. A rainbow scale was used to show the fluorescence intensity of the Ca^2+^ indicator (blue: low values; red: high values); and (**D**) representative trace of changes over time in nuclear Ca^2+^ levels recorded in a neuron transfected with GCaMP3-NLS before (baseline) and after the addition of 500 nM AβOs at the indicated time (arrow). Data are expressed as representative recordings from experiments performed with different hippocampal cultures; *n* = 3.

**Figure 2 antioxidants-12-01972-f002:**
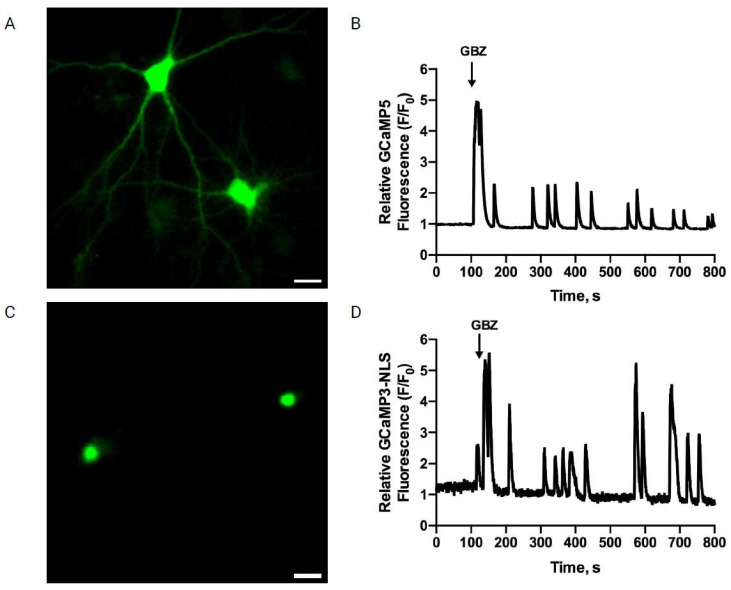
Gabazine addition to primary hippocampal neurons induces synchronized cytoplasmic (**A**,**B**) and nuclear Ca^2+^ signals (**C**,**D**). Gabazine induces synchronized cytoplasmic and nuclear Ca^2+^ signals in primary hippocampal neurons: (**A**) representative image showing the fluorescence signals emitted by the cytoplasmic Ca^2+^ sensor GCaMP5 (green); (**B**) representative trace showing the relative changes of the GCaMP5 sensor fluorescence signals recorded from the dendrites of neurons before (baseline) and after the addition of 5 μM GBZ to the culture, which caused synchronous oscillatory Ca^2+^ transients; (**C**) representative image showing the fluorescence signals emitted by the nuclear Ca^2+^ sensor GCaMP3-NLS (green) recorded in basal condition; two nuclei of hippocampal neurons are observed; and (**D**) traces of recordings of the relative changes of the signal intensity of GCaMP3-NLS, recorded in the nuclei illustrated in D before (baseline) and after the addition of 5 μM GBZ to a control culture at the indicated time (arrow). **A** and **C**, Scale bar: 20 μm. Data are expressed as representative recordings from experiments performed with different hippocampal cultures; *n* = 3.

**Figure 3 antioxidants-12-01972-f003:**
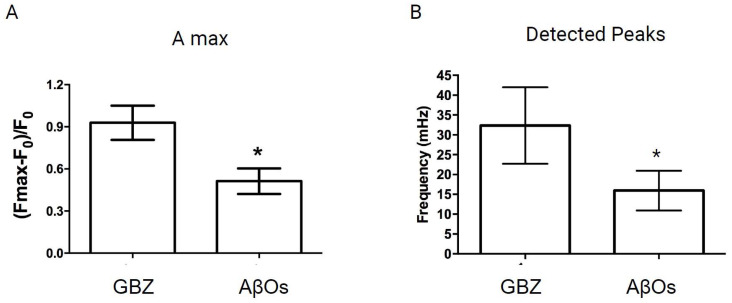
Treatment with AβOs disrupts the transient nuclear Ca^2+^ signals induced by GBZ in pyramidal hippocampal neurons. Treatment with AβOs disrupts the transient nuclear Ca^2+^ signals induced by GBZ in pyramidal hippocampal neurons. Neurons were incubated for 6 h with AβOs (500 nM) before GBZ addition and the relative change in fluorescence with time values (F_max_ − F_0_/F_0_) was recorded in the nucleus of neurons expressing GCaMP3-NLS. (**A**) The maximum amplitude (GBZ mean value = 0.95; AβOs mean value = 0.30) and (**B**) the frequency of the detected peaks were quantified (GBZ mean value = 38.5; AβOs mean value = 15.83); the average values obtained from five experiments performed in independent cultures show significant differences between GBZ addition to controls or to neurons preincubated with AβOs. Data are presented as Mean ± SE. In total, at least 50 neurons were analyzed per condition in five independent experiments (with different primary cultures). Statistical analysis was conducted using Student’s *t*-test. In (**A**,**B**), * *p* < 0.05; the exact *p* values were *p* = 0.040 and *p* = 0.037, respectively.

**Figure 4 antioxidants-12-01972-f004:**
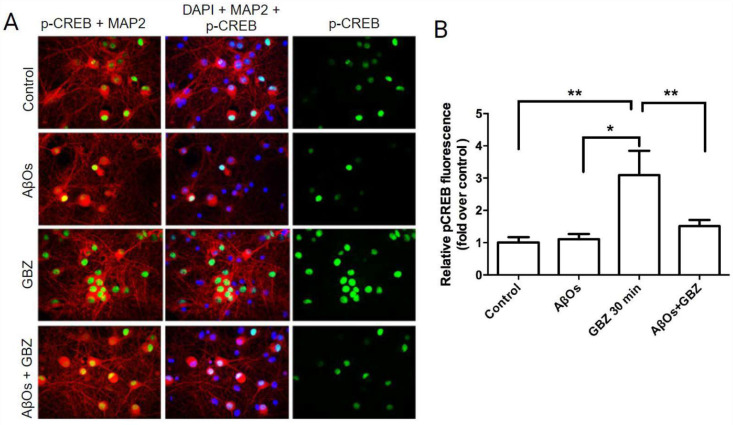
Treatment with AβOs prevents the CREB phosphorylation increase induced by GBZ in hippocampal neurons: (**A**) representative images of control hippocampal cultures and/or cultures pre-incubated for 6 h with 500 nM AβOs before the addition of GBZ. The images show that the signals originated from incubation with antibodies against phosphorylated CREB (p-CREB; green) and the neuronal marker MAP2 (red); cell nuclei were labeled with DAPI (blue). All images illustrate the sum of the fluorescence recorded in the *z*-axis of all confocal planes acquired by spinning-disk microscopy, with a 40× objective, scale bar 20 μm; and (**B**) quantification of the relative levels of p-CREB fluorescence detected in different conditions (AβOs mean value = 1.106; GBZ mean value = 3.093, and AβOs + GBZ mean value = 1.510). Bars represent Mean ± SE, *n* = 3. Statistically significant differences were evaluated by one-way ANOVA followed by Bonferroni’s post hoc test for multiple comparisons. * *p* < 0.05; ** *p* < 0.01 The exact *p* values from left to right are as follows: *p* = 0.003, *p* = 0.030, and *p* = 0.005.

**Figure 5 antioxidants-12-01972-f005:**
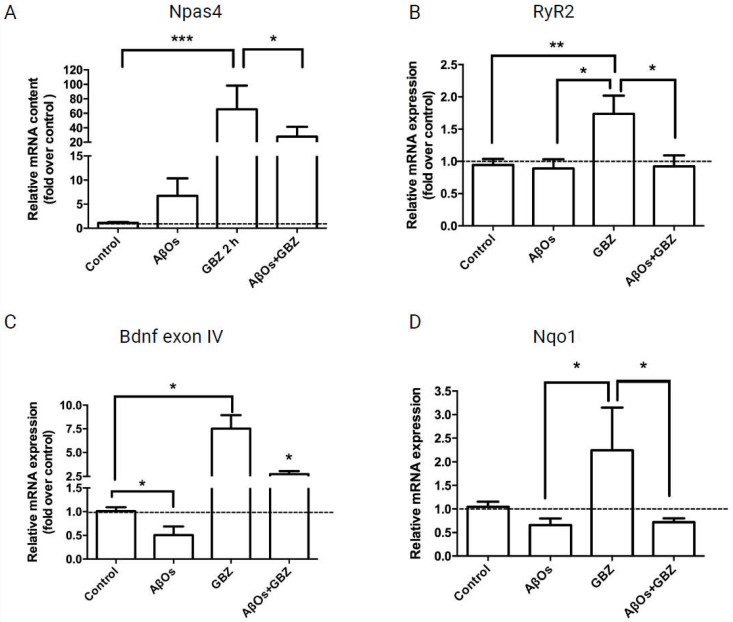
Treatment with AβOs decreases or prevents the gabazine-induced increase in the mRNA levels of Npas4, RyR2, BDNF exon IV and the antioxidant enzyme Nqo1. The relative mRNA levels of Npas4 (mean fold change value from left to right 7.065; 91.43; 24.83) (**A**); RyR2 (mean fold change value from left to right 0.891; 1.736; 0.924) (**B**); BDNF exon IV (mean fold change value from left to right 0.507; 7.519; 2.753) (**C**); and Nqo1 (mean fold change value from left to right 0.572; 3.149; 0.668) (**D**) were determined by qPCR of primary neuronal cultures treated with GBZ for 2 h and incubated next for 6 h with 500 nM AβOs or saline. Values, normalized for β-actin mRNA levels, are expressed as fold over the values displayed by control cultures. Data are expressed as Mean ± SE; *n* = 5. Statistical analysis was performed with ANOVA followed by Tukey’s post hoc test for multiple comparisons. * *p* < 0.05; ** *p* < 0.01; *** *p* < 0.001. The exact *p* values from left to right are as follows *p* = 0.0002 and *p* = 0.0311 for (**A**); *p* = 0.0090; *p* = 0.0130 and *p* = 0.0246 for (**B**); *p* < 0.001; *p* < 0.001; *p* = 0.0391 for (**C**); *p* = 0.0133; and *p* = 0.0391 for (**D**).

## Data Availability

The data that support the findings of this study are available upon request from the authors.

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
