# Peer review of "Amyloid β-Oligomers Inhibit the Nuclear Ca2+ Signals and the Neuroprotective Gene Expression Induced by Gabazine in Hippocampal Neurons"

_antioxidants, 2023, doi:10.3390/antiox12111972_

Round 1

Reviewer 1 Report

Comments and Suggestions for Authors

There is a contradiction between the two statements of the authors. 

Page 6.

row 243 and 244, and the statement at row 269 and 270.

"Previous reports have described that addition of AβOs elicits long-lasting and low ampli- tude cytoplasmic Ca2+ signals in primary hippocampal neurons [12,13]. Following AβOs  addition (500 nM), cytoplasmic Ca2+ signals, detected with Fluo-4 (Figure 1A), exhibited a 245 similar response pattern (Figure 1B). "

.."treatment with 500 nM AβOs (6 h) did not modify the ba- 269 sal cytoplasmic Ca2+ levels (Supplementary Figure 1). "

Can the authors explain the two differences?

Author Response

We thank the reviewer for indicating that this point was not clearly described in our manuscript. However, the two statements are not contradictory. The statement made in our first version in page 6, rows 243-244, describes the results of a previous study made by our laboratory, and confirmed in the present work, in which we found that AβOs addition elicits long-lasting and low amplitude cytoplasmic Ca2+ signals in primary hippocampal neurons, which last for several minutes. These results show that AβOs cause a transient increase in Ca2+ levels. On the other hand, the statement presented in the revised version highlights that basal cytoplasmic Ca2+ levels were not modified following treatment for 6 h with 500 nM AβOs. To explain these differences, we added the following sentences (lines 309-313): “Of note, treatment with 500 nM AβOs (6 h) did not modify the basal cytoplasmic Ca2+ levels (Supplementary Figure 2). These results indicate that the moderate increase in cytoplasmic Ca2+ levels initially caused by AßOs, which lasts for at least 30 minutes (Paula-Lima et al, 2011), did not persist in primary hippocampal neurons incubated for 6 h with 500 nM AβOs.

Reviewer 2 Report

Comments and Suggestions for Authors

The manuscript by Lobos and collaborators describes how Amyloid-beta oligomers (AβOs) inhibit the generation of nuclear Ca 2+ signal induced by Gabazine (GBZ) addiction and further reduce the expression of genes engaged in synaptic plasticity and the antioxidant response. To realize their scopes, the authors used primary rat hippocampal cultures. For most of the experiments, hippocampal neurons were pre-incubated with 500 nM AβOs for 6 h before incubation with 5 μM GBZ for 30 minutes for Ca2+ measurements and CREB immunofluorescence; for determinations by qPCR, cultures were incubated for 6 h with 500 nM AβOs before incubation with 5 μM GBZ for 2 h. Compounds were maintained during the respective incubation period as stated. 

The manuscript is very clear regarding the content and the message it wants to send, i.e. how beta oligomers interfere with calcium signaling both at the cytoplasmic and nuclear level. Unfortunately, the results are not sufficient to demonstrate the thesis.

- It does not present a dose-response of GBZ on hippocampal neurons per se and in the presence of 500nM AβOs. Without these data, all the findings do not show an appreciable value.

- The use of Gabazine also called SR-95531, must be compared with one or more other GABA A selective antagonists (i.e. Bicucilline, furosemide) or GABA generic antagonist (i.e. Picrotoxin). 

- To demonstrate the effective role of GBZ in nuclear Ca 2+ signal induction must be used a specific GABA agonist (i.e. Zonisamide).

- Moreover, the work should be completed by an electrophysiological recording proving the lack of synaptic plasticity after treatment or when genes are not active.

- Should be indicated to perform 2 or 3 points of incubation with  AβOs (i.e. 2/6/12 hr). To compare the effective activation in response to the time of action AβOs, mimicking different levels of protein precipitation in humans.

- Concerning neuroinflammation, it is strongly suggested to show a representative plot and correlate CD8+ infiltration with the related alterations of Npas4 and Arc via biochemical assays (Western Blotting or Flow Cytometry).

The title should be changed to” Role of GBZ in nuclear Ca 2+ signal: AβO destroys Calcium-mediated Gene Expression in hippocampal neurons.”

-  Conclusion: I would like the authors to extend the conclusions more, emphasizing novelty and mentioning evidence in the literature of using Amyloid-beta oligomers as possible therapeutic targets. Are there any clinical trials? Is there evidence in the animal model? I find the conclusions too brief. Perhaps it would also be better to devote a small paragraph to the prospects, following the results collected by the authors.

- I suggest reading and citing this work in the conclusion paragraph: https://doi.org/10.1016/j.ijbiomac.2023.124231.

Minor point:

- Check the quality of the Graphical Abstract, it appears grainy. The Graphical Abstract should be a high-quality illustration or diagram in any of the following formats: PNG, JPEG, or TIFF. The minimum required size for the GA is 560 × 1100 pixels (height × width). The size should be of high quality to reproduce well.

- I suggest adding the keywords: Hippocampal cultures Amyloid-beta oligomers Gabazine

-     The introduction is well structured, places the study briefly in a broad context, and defines the purpose of the work and its significance. The current state of the research field is examined. However, the last part of the introduction should highlight the novelty of this work.

- I encourage the authors to cite these articles in the introduction, which may provide fascinating insights into their research:

https://doi.org/10.1007/s00018-023-04876-8

doi: 10.14348/molcells.2023.2156. Epub 2023

- Material and Methods describe sufficient detail to allow others to replicate and build on published results. The codes and companies of the reagents and materials used are given.

- Authors should pay attention to line 139: the reference (Paula-Lima et al. 2021) should be entered in the correct format provided by the guidelines provided by the journal. If not cited, first recheck the correct order of the references.

- Results: I wondered why the authors focused only on the mRNA expression levels of BDNF exon 4.

- For example, the sequence (5′-TCACGTCA-3′) located 35 bp from a known transcription start site of exon 3 (Timmusk et al. 1993) has been shown to mediate l-type Ca2+-dependent BDNF gene expression via CREB or some other related transcription factor in primary embryonic cortical neurons (Tao et al. 1998).

- For mRNA expression analysis of Npas4, RyR2, BDNF exon IV, and Nqo1, how many n were used? The same for the evaluation of CREB, n=3?

- I advise the authors to also report the results and not just the legends, the statistics, averages, and SEM, Pvalue of the data obtained.

- Please add ABOligomers abbreviation

Author Response

General Comment: The manuscript by Lobos and collaborators describes how Amyloid-beta oligomers (AβOs) inhibit the generation of nuclear Ca2+ signal induced by Gabazine (GBZ) addiction and further reduce the expression of genes engaged in synaptic plasticity and the antioxidant response. To realize their scopes, the authors used primary rat hippocampal cultures. For most of the experiments, hippocampal neurons were pre-incubated with 500 nM AβOs for 6 h before incubation with 5 μM GBZ for 30 minutes for Ca2+ measurements and CREB immunofluorescence; for determinations by qPCR, cultures were incubated for 6 h with 500 nM AβOs before incubation with 5 μM GBZ for 2 h. Compounds were maintained during the respective incubation period as stated. The manuscript is very clear regarding the content and the message it wants to send, i.e. how beta oligomers interfere with calcium signaling both at the cytoplasmic and nuclear level. Unfortunately, the results are not sufficient to demonstrate the thesis.

Answer: Thanks for your comment. We used a concentration of 5 µM gabazine (GBZ) to effectively block GABA(A) receptors, which has been validated by several previous reports. For example, Ueno et al., 1997 (https://doi.org/10.1523%2FJNEUROSCI.17-02-00625.1997), compared the inhibitory effects of GBZ with those of other GABA(A) receptor antagonists, such as Bicuculline, pentobarbital, and alphaxalone, and concluded that 5-10 µM GBZ effectively blocks neuronal GABA(A) receptors. Additionally, GBZ is considered a more potent and specific GABA(A) receptor antagonist than Bicuculline (Johnston, 2013; https://doi.org/10.1111/bph.12127). It is also worth mentioning that 3-10 μM GBZ blocks GABA(A) receptors in primary hippocampal neurons (Sipila et al., 2005;10.1523/JNEUROSCI.0378-05.2005). Furthermore, we previously reported (Lobos et al., 2021, doi:10.1073/pnas.2102265118) that incubation of primary hippocampal neurons with 5-10 µM GBZ induces neuronal activity, promotes the emergence of cytoplasmic and nuclear calcium signals, and increases CREB phosphorylation and the expression of genes involved in synaptic plasticity. In light of this, we added the following text to the Discussion section (Lines 381-388): “In this work, we treated primary hippocampal cultures with 5 µM GBZ to induce neuronal activity, which effectively inhibits GABA(A) receptors (Ueno et al., 1997; Sipila et al., 2005; Lobos et al., 2021). Moreover, GBZ is a more potent and specific GABA(A) receptor antagonist than Bicuculline, pentobarbital, and alfaxalone (Ueno et al., 1997; Johnston, 2013). Additionally, we recently reported (Lobos et al., 2021) that the addition of 5-10 µM GBZ to primary hippocampal neurons induces neuronal activity, promotes the emergence of cytoplasmic and nuclear calcium signals, and increases CREB phosphorylation and the expression of genes involved in synaptic plasticity and neuroprotection.”

Comment 3: To demonstrate the effective role of GBZ in nuclear Ca2+ signal induction must be used a specific GABA agonist (i.e. Zonisamide).

Answer: Thank you for your comment. However, the current article does not aim to demonstrate the effective role of GBZ, in fact considering that the specific GABA receptor agonist Zonisamide inhibits action potential bursting, and consequently the Ca2+ signals elicited by neuronal activity . Hence, we cannot use Zonisamide, since our current research focuses on investigating the role of AβOs on the Ca2+ signaling pathways induced by GBZ, which by inhibiting GABAergic transmission promotes neuronal activation by inhibition of inhibitory synapses. In addition, we reported the effective role of GBZ on nuclear calcium signal induction in a previous report, published in Proceedings of the National Academy of Sciences (Lobos et al., 2021; doi:10.1073/pnas.2102265118).

Comment 4: Moreover, the work should be completed by an electrophysiological recording proving the lack of synaptic plasticity after treatment or when genes are not active.

Answer: Thank you for your comment. Performing electrophysiological recordings and monitoring the expression of synaptic genes are beyond the scope of the present work. However, we appreciate your input and will consider the possibility of further experiments in this direction, since electrophysiological recordings coupled with the expression of genes engaged in synaptic plasticity would provide valuable insights into the mechanisms of AβOs-invoked disruption of hippocampal function. However, previous reports have consistently shown that AβOs are “synaptotoxins”, which lead to cognitive impairment in Alzheimers disease. Thus, in normal rodent hippocampus AβOs inhibit synaptic plasticity (Shankar et al., 2008; https://doi.org/10.1038/nm1782), via their inhibitory effects on long-term potentiation (LTP), their enhancement of long-term depression (LTD), and their inhibition of structural plasticity via reducing dendritic spine density. Nevertheless, and in consideration of your comment, we added the following paragraph to the Discussion section (Lines 389-398): “Previous studies have reported that AβOs are synaptotoxins that lead to cognitive impairment in AD [68] and impede synaptic plasticity by inhibiting long-term potentiation (LTP), enhancing long-term depression (LTD) [69]; AβOs also impair structural plasticity by reducing dendritic spine density in normal rodent hippocampus and primary hippocampal neurons [12,69]. Electrophysiological recordings coupled with the expression of synaptic genes would provide valuable insights into the mechanisms of AβOs-invoked disruption of synaptic plasticity induced by GBZ. Our findings on the inhibition of Ca2+signaling produced by AβOs in primary hippocampal neurons provide a foundation for future research exploring synaptic plasticity and electrophysiological aspects in more detail.

.”

Comment 5: Should be indicated to perform 2 or 3 points of incubation with AβOs (i.e. 2/6/12 hr). To compare the effective activation in response to the time of action AβOs, mimicking different levels of protein precipitation in humans.

Answer: Thank you for this suggestion, we have now included an experiment as Supplementary Figure 2, which shows that 15 minutes of incubation with AβOs is not sufficient to inhibit the effects of GBZ, an indication that the inhibitory effects of AβOs require longer incubation times. The revised text now reads (Lines 295-299):” Primary cultures were incubated for 6 h with 500 nM AβOs since this protocol significantly alters hippocampal function by deregulating glutamate neurotransmission, leading to synapse failure [29]. As depicted in Supplementary Figure 2, 15 minutes incubation with AβOs did not inhibit the induction of nuclear Ca2+ signals by GBZ”.

Comment 6: Concerning neuroinflammation, it is strongly suggested to show a representative plot and correlate CD8+ infiltration with the related alterations of Npas4 and Arc via biochemical assays (Western Blotting or Flow Cytometry).

Answer: Thank you.  However, the primary focus of our study was to evaluate the impact of AβOs on GBZ-induced Ca2+signals and the expression of genes engaged in synaptic plasticity and neuroprotection. Thus, although there is a strong connection between neuroinflammation and neurodegenerative processes, including the infiltration of CD8+ T lymphocytes and their influence on gene expressions related to synaptic plasticity, these studies which require an in vivo setting are beyond the scope of the present work.  Nevertheless, we included this aspect in the Discussion section because neuroinflammation has been reported to play a role in the regulation of the excitatory/inhibitory tone in the brain. The text now reads (lines 461-463):” Moreover, our results, which indicate that AβOs reduced the increase in Npas4 gene expression induced by gabazine in primary hippocampal neurons, open a new direction for extensive future research, including neuroinflammation.”

Comment 7: The title should be changed to” Role of GBZ in nuclear Ca 2+ signal: AβO destroys Calcium-mediated Gene Expression in hippocampal neurons.

Answer: We thank the reviewer for this suggestion, since we understand that it aims at specifying that we have results on GBZ-induced neuronal activity. Hence, we replaced the title of our manuscript, which now reads: “Amyloid β-oligomers inhibit the nuclear Ca2+ signals and the neuroprotective gene expression induced by gabazine in primary hippocampal neurons.”

Comment 8: Conclusion: I would like the authors to extend the conclusions more, emphasizing novelty and mentioning evidence in the literature of using Amyloid-beta oligomers as possible therapeutic targets. Are there any clinical trials? Is there evidence in the animal model? I find the conclusions too brief. Perhaps it would also be better to devote a small paragraph to the prospects, following the results collected by the authors.

Answer: Thank you for your thoughtful comments. We extended our conclusions in the revised version (Lines 511-529), which now read: “Based on the results of this study, which provides the first integrative evidence that AβOs disrupt the generation of activity dependent nuclear Ca2+ signals and reduce the expression of genes engaged in synaptic plasticity and the antioxidant response, we propose that these disruptions are likely to contribute to the cognitive decline and the neurodegeneration that characterize Alzheimer's disease. Moreover, this work adds to the current understanding of the toxic effects of AβOs on hippocampal neuronal function and reveals new possible therapeutic targets to treat AD. Hence, one potential therapeutic approach to slow or prevent AD progression would be to restore activity induced nuclear Ca2+signaling and early gene expression. More research is needed to develop and test these therapeutic approaches. Of note, a recent report analyzes how AβOs can become potential therapeutic targets for AD [107].  Recently several clinical trials using monoclonal antibodies that target amyloid plaques or AβOs in early AD have shown positive results and even obtained approval, although with important clinical efficacy and safety issues [108-110]. In order to overcome these problems, it is crucial to first gain a deeper understanding of the mechanisms that come into play and ultimately disrupt the regulation of neuronal activity induced by AβOs, with significant implications for tackling the heterogeneity of this devasting pathology and for developing diagnoses and novel pharmacological targets.”

Comment 9: I suggest reading and citing this work in the conclusion paragraph: https://doi.org/10.1016/j.ijbiomac.2023.124231.

Answer: Thank you for your suggestion. We now cite the paper by Ghosh et al., 20203, in the conclusion paragraph (line 522) of the revised version.

Minor points:

Comment 10: Check the quality of the Graphical Abstract, it appears grainy. The Graphical Abstract should be a high-quality illustration or diagram in any of the following formats: PNG, JPEG, or TIFF. The minimum required size for the GA is 560 × 1100 pixels (height × width). The size should be of high quality to reproduce well.

Answer: Thanks for the thorough revision. We now added the graphical abstract as JPG 600 DPI (5025 x 3518 pixels).

Comment 11: I suggest adding the keywords: Hippocampal cultures Amyloid-beta oligomers Gabazine

Answer: Thank you. We now added these keywords.

Comment 12: The introduction is well structured, places the study briefly in a broad context, and defines the purpose of the work and its significance. The current state of the research field is examined. However, the last part of the introduction should highlight the novelty of this work.

Answer:  In order to highlight the novelty of our work, we now added in the introduction section the following sentences, (Lines 130-138): “In this study, by revealing that AβOs disrupt crucial nuclear Ca2+ signaling processes and suppress the expression of genes associated with neuroprotection in primary hippocampal cultures, we present novel results that advance our current knowledge of the noxious effects of AβOs on rodent hippocampal function. Hence, this work provides a framework for a deeper understanding of how AD-related mechanisms impact synaptic plasticity and neuronal health. Moreover, it offers promising avenues for therapeutic interventions in AD and related neurodegenerative conditions.”

Comment 13: I encourage the authors to cite these articles in the introduction, which may provide fascinating insights into their research https://doi.org/10.1007/s00018-023-04876-8; doi: 10.14348/molcells.2023.2156.

Answer: Thank you for your suggestion. We cited the articles you suggest in the introduction (Lines 85-92): “Of note, it was recently shown in hippocampal neurons that steep increases in nuclear Ca²⁺ levels induce instantaneous uncoupling of a protein called Jacob from LaminB1 at the nuclear lamina and promotes the association of Jacob with the transcription factor cAMP-responsive element-binding protein (CREB).  (Karpova et al., 2023). Interestingly, transient receptor potential V1 (TRPV1) deficiency, by promoting the BDNF/TrkB signaling pathway, prevents hippocampal cell death in a 3xTg-AD mice (Kim et al., 2023).”

.

Comment 13: Material and Methods describe sufficient detail to allow others to replicate and build on published results. The codes and companies of the reagents and materials used are given.

Answer: Thank you for your comment.

Comment 14: Authors should pay attention to line 139: the reference (Paula-Lima et al. 2021) should be entered in the correct format provided by the guidelines provided by the journal. If not cited, first recheck the correct order of the references.

Answer: Thank you for your detailed revision. We revised the reference and we now cite it in the correct format.

Comment 15: Results: I wondered why the authors focused only on the mRNA expression levels of BDNF exon 4. For example, the sequence (5′-TCACGTCA-3′) located 35 bp from a known transcription start site of exon 3 (Timmusk et al. 1993) has been shown to mediate l-type Ca2+-dependent BDNF gene expression via CREB or some other related transcription factor in primary embryonic cortical neurons (Tao et al. 1998).

Answer: Thank you.  We focused on exon 4 because it encodes the mature BDNF protein strongly expressed in hippocampus by neuronal activity as described in recent reports now cited.  Hence, by focusing on the mRNA expression levels of exon 4, we can get a better estimate of the levels of mature BDNF protein in the cell induced by neuronal activity and calcium signaling. Also, the expression of exon 4 is more responsive to changes in synaptic activity. Recent studies have shown that the mRNA expression of exon 4 is more rapidly and strongly upregulated in response to synaptic activity than the mRNA expression of exon 3 (Esvald et al., 2022; https://doi.org/10.1523/JNEUROSCI.2535-21.2022); therefore, changes in exon 4 expression may be a more sensitive indicator of synaptic plasticity than changes in exon 3 expression. We added the following sentence to the Discussion section (Lines 483-490): “In this work, we focused on the expression of BDNF exon 4 transcript because it encodes the mature BDNF protein, it is broadly expressed and strongly induced by neuronal activity in cortical and hippocampal neurons, with a tight temporal, spatial, and stimulus-specific regulation compared to mRNA expression of other BDNF transcripts like exon 3; of special interest, several calcium-response elements are found within the Bdnf promoter IV region, and CREB in combination with its coactivators CBP and CRTC1 specifically regulate its early induction in hippocampal neurons [94, 95].”

Comment 16: For mRNA expression analysis of Npas4, RyR2, BDNF exon IV, and Nqo1, how many n were used? The same for the evaluation of CREB, n=3?

Answer: Thank you. We used an N≥5 for each gene. We now added both in the Results section and the Legends the statistics, the average values, the SEM and the P-values of all the data presented.

Comment 17: I advise the authors to also report the results and not just the legends, the statistics, averages, and SEM, Pvalue of the data obtained.

Answer: As described in our preceding answer, we now added in the Results section the statistics, the SE and the P-values of all data presented.

Comment: Please add ABOligomers abbreviation

Answer: We now added an abbreviation section containing this and the other abbreviations used in the text (Lines 670-677).

Round 2

Reviewer 2 Report

Comments and Suggestions for Authors

publish in present form